# Ten Years of Deformed Wing Virus (DWV) in Hawaiian Honey Bees (*Apis mellifera*), the Dominant DWV-A Variant Is Potentially Being Replaced by Variants with a DWV-B Coding Sequence

**DOI:** 10.3390/v13060969

**Published:** 2021-05-24

**Authors:** Isobel Grindrod, Jessica L. Kevill, Ethel M. Villalobos, Declan C. Schroeder, Stephen John Martin

**Affiliations:** 1School of Environment and Life Sciences, University of Salford, Manchester M5 4WX, UK; i.r.grindrod@edu.salford.ac.uk; 2Veterinary Population Medicine, College of Veterinary Medicine, University of Minnesota, St. Paul, MN 55108, USA; J.kevill@bangor.ac.uk; 3School of Natural Sciences, Bangor University, Bangor, Gwynedd LL57 2UW, UK; dcschroe@umn.edu; 4College of Tropical Agriculture and Human Resources, University of Hawaii at Mānoa, 3050 Maile Way, Honolulu, HI 96822, USA; emv@hawaii.edu; 5Environmental Biology (Virology), School of Biological Sciences, University of Reading, Reading RG6 6AH, UK

**Keywords:** honey bee, deformed wing virus, *Varroa*

## Abstract

The combination of Deformed wing virus (DWV) and *Varroa* *destructor* is arguably one of the greatest threats currently facing western honey bees, *Apis mellifera*. *Varroa’*s association with DWV has decreased viral diversity and increased loads of DWV within honey bee populations. Nowhere has this been better studied than in Hawaii, where the arrival of *Varroa* progressively led to the dominance of the single master variant (DWV-A) on both mite-infested Hawaiian Islands of Oahu and Big Island. Now, exactly 10 years following the original study, we find that the DWV population has changed once again, with variants containing the *RdRp* coding sequence pertaining to the master variant B beginning to co-dominate alongside variants with the DWV-A *RdRp* sequence on the mite-infested islands of Oahu and Big Island. In speculation, based on other studies, it appears this could represent a stage in the journey towards the complete dominance of DWV-B, a variant that appears better adapted to be transmitted within honey bee colonies.

## 1. Introduction 

Western Honey bees (*Apis mellifera*) and the pollination services they provide are important both economically and environmentally [1]. However, concerns for the health of honey bee populations have been mounting over the years as they face a whole host of threats, including pollution, pests, and parasites [2,3,4]. No single threat can be isolated as the leading factor, but the bee-mite-virus tripartite relationship is an integral part of this struggle. The ectoparasite mite *Varroa destructor*, first became a problem around the 1940s when it jumped species from Eastern (*Apis cerana*) to Western honey bees and was traded across the globe [5]. Being naïve to this new threat, *A. mellifera* populations were easily overwhelmed and collapsed. Whilst *Varroa* can directly weaken honey bee adults and pupae, their true lethality lies in their ability to vector the Deformed wing virus (DWV). 

Prior to the spread of *Varroa,* DWV, originally known as the Egyptian bee virus, was known only from a few rare cases [6]. Indeed, despite its long co-existence with honey bees, it was only isolated in 1986 [7]. This is largely because, without *Varroa,* DWV was limited to less effective oral and sexual transmission routes, and as a consequence, it existed at low viral loads as a covert and usually symptomless infection [8,9]. DWV only became a major problem for honey bees after *Varroa* arrived and, through its feeding habits, introduced a new, highly effective transmission mechanism [8]. This direct injection of DWV causes emerging adults to have a shortened abdomen, a reduced lifespan [10], precocious foraging [11,12], and if the virus happens to replicate in the wing buds of the pupae, deformed wings [8]. If infection rates are high, the reduced longevity quickly leads to an imbalanced workforce and a collapsing of the colony, particularly during the winter period for bees in the northern hemisphere. Precocious foraging, which DWV can stimulate, accelerates the behavioral and physiological maturation of worker bees, further reducing their lifespan [12].

Accordingly, in areas without DWV, such as Papua New Guinea, Solomon Islands [13], colonies are able to tolerate *Varroa* without suffering colony losses. Similarly, in areas absent of *Varroa,* colonies do not succumb to DWV infections, as genome equivalents are very low and highly diverse [9]. A pivotal study in Hawaii found that prior to the spread of *Varroa,* DWV infections consisted of a diverse array of variants, and post *Varroa*, this diversity was drastically reduced [9], a finding that was independently found in the UK honey bees [14]. This variant called DWV-A is one of the three highly successful variants, known as master variants, which make up the DWV quasispecies [15]. DWV-A includes the classical versions of DWV and Kakugo virus. The other two master variants are DWV-B, previously known as Varroa destructor virus 1 (VDV-1), and DWV-C, which is the rarest of the three [16]. Within quasispecies, the master variants exist surrounded by a ‘cloud’ of less successful variants that are generated due to the rapid mutation of the RNA genome [15].

The transmission pathway introduced by *Varroa* has altered the dynamics of the quasi-species by favoring particular variants that can survive within the bee [14,17], and now can replicate within mites’ salivary glands [18], be efficiently transmitted by mite feeding [14], and replicate to high levels within the bee [14]. Originally only the master variant DWV-A was detected, and this was associated with the death of infested colonies, later another dominant variant DWV-B appeared [19]. Large scale surveys and longitudinal studies are showing that where DWV and *Varroa* are present, DWV-A and DWV-B seemingly vie for dominance, with a pattern of the increasing dominance of DWV-B [16,20]. This change could possibly be explained by DWV-B, unlike DWV-A, which can replicate within the mite [18]. Furthermore, co-infection with more than one DWV variant has led to the identification of DWV recombinant genomes [21,22,23,24]. To date, several recombinants have been detected in honey bees, between DWV-A and DWV-B [22,23,24], and also DWV-A and DWV-C [21]. The most commonly detect recombinant breakpoints have been located in the *5′UTR* [23], *Lp*, *Vp1*, *Vp2*, *Vp3*, *helicase* [22], and more recently, a recombinant between DWV-A and an unknown variant in the *Vpg* and RNA dependent RNA polymerase (*RdRp*) coding sequences [24].

In 2012, DWV-B was first detected in samples from *Varroa*-infested Hawaiian Islands [25] and again in 2016 [26]. Therefore, 10 years on from the original Hawaiian study that sampled 239 colonies detecting primarily DWV-A [9], we returned to resample three island populations. Here, we investigate how DWV has changed in respect to prevalence and load of DWV-A and -B *RdRp* coding sequence, a highly conserved region of the genome, and then compare any changes to the current global status of DWV. During the past 10 years, the *Varroa* status of the Hawaiian Islands has remained the same with Maui and Kauai been mite-free while *Varroa* is ubiquitous on Oahu and Big Island, where colonies are treated with miticides regularly, although a small number of beekeepers are maintaining increasing numbers of colonies without treating [27].

## 2. Methods

### 2.1. Sample Collection

Samples were collected during November 2019, 10 years after the original collection date in the field (Nov 2009 and 2010), and stored on ice before being transferred into ethanol for storage at −20 °C. Samples of at least 30 adult bees were collected from both the *Varroa* infested islands of Oahu (*n* = 41 colonies, *n* = 6 apiaries, *n* = 11 feral colonies), Big Island (*n* = 43 colonies, *n* = 9 apiaries, *n* = 1 feral colony), and the *Varroa*-free island of Kauai (*n* = 22 colonies, *n* = 4 apiaries, *n* = 2 feral colonies). Two of the 11 feral samples on Oahu, T4 and UH127, only 29 bees were collected from each colony.

In addition, 2 sets of 5 pupal samples were taken from 2 colonies on Oahu from an apiary that showed the signs of natural-mite resistance. All samples were transported directly too and processed 1–2 months later at the University of Minnesota.

### 2.2. Sample Processing

For each sample: 30 asymptomatic bees were dried of residual ethanol and individually inspected for *Varroa*, and if present, the mite was removed. This was to prevent contamination of the samples with viral RNA from *Varroa* and to standardize the test. The bees were frozen with liquid nitrogen and homogenized by a mill mixer (Ritesch) for 30 s. The Oahu pupal samples were also inspected for *Varroa* and if present, any mites were removed. The pupae were individually dried, frozen with liquid nitrogen, and crushed within an Eppendorf tube using a sterile pipette tip. The bee material was then stored at −80 °C until RNA extraction. An empty open Eppendorf tube served as a blank for any aerial contamination during the crushing process.

### 2.3. RNA Extraction and Quantification

RNA was extracted from the 50 mg of each sample using the MagMAX mirVana total RNA isolation kit with the MagMAX express 96 on program AM1830_DW (Applied Biosystems, Foster City, CA, USA). Following the manufacturer’s protocol, 302.1 µL of lysis binding mix (300 µL of lysis buffer and 2.1 µL of 2-Mercaptoethanol) was added to each sample, and the samples were vortexed for 15 s before being put into the 5× *g* for 5 min at 2000 rpm. The manufacturer’s protocol was modified slightly, thus 150 µL of the lysate was put into each well of the processing plate rather than 100 µL. To each sample on this plate, 20 µL of binding mix (10 µL RNA beads and 10 µL enhancer) was added, and the plate shook for 5 min using the plate shaker Lab-Line™ at 950 rpm.

In total, RNA was extracted from 116 samples, 9 blanks from the crushing stage and 2 negatives to check for contamination during the extraction process. RNA was quantified using the Nanodrop 2000 (Thermo Fisher Scientific, Waltham, MA, USA) and standardized to 50 ng/µL per sample using RNase free water before storage at −80 °C.

### 2.4. RT -qPCR

To quantify the viral load of each DWV master variant, RT-qPCR was performed on the 116 samples using the ABC assay method [28]. The samples were screened for the DWV master variants A, B, and C, using primers targeting the *RdRp* region and, therefore, this assay can only provide insight into the presence of each DWV master variants and associated recombinants at the time of sampling. It cannot report on the prevalence of any DWV recombinant but rather provides an overview of whether there was a shift from DWV-A and its associated recombinants and DWV-B and its associated recombinants using a conserved region of the viral genome.

Reactions were performed on a quant studio 3 (Applied Biosystems/Thermo Fisher Scientific, USA), using a powerup SYBER^®^ Green RNA-to-Ct 1-Step kit^TM^ from applied Biosystems. The 50 ng/µL samples were run singly alongside a 10-fold dilution series run in triplicate. The 10-fold dilution series was made using a standard specific to each DWV master variant, the concentration of which was determined using the Nanodrop 2000 (Thermo Fisher Scientific) before dilution. Reactions contained 1 µL of the 50 ng/µL RNA sample and 9 µL of master mix. The master mix was comprised of 0.08 µL reverse transcriptase, 1 µL DWV forward primer and 1 µL DWV reverse primer (Type A, B or C), 5 µL PCR mix, and 1.92 µL H_2_O. A negative control consisting of 1 µL H_2_O and 9 µL master mix was included on each PCR plate. An actin control was not deemed necessary as the samples had not undergone long-term storage. The reactions were run on the quant studio 3, the reverse transcription stage occurred at 45 °C for 10 min and denaturation at 95 °C for 10 min, followed by 35 cycles of denaturation at 95 °C for 15 s, annealing at 58 °C (types A and B) or 61 °C (type C) for 15 s and extension at 72 °C for 15 s. The final stage was a dissociation melt curve at 70 to 95 °C, this was to check for any contamination.

### 2.5. Analyzing the Results

DWV-C was not detected in the screened samples; therefore, results were analyzed for DWV-A and -B only. The average viral copy number was calculated by the quantstudio software. The average viral copy number was used to calculate the quantity DWV genome equivalent per bee. This was obtained using the formula:Genome equivalent = (average copy number) × (RNA dilution factor) × (elution volume of RNA) × (proportion of bee material)

The dilution factor can be calculated by dividing the RNA concentration of the original sample (before it was diluted) by 50 (the concentration it was diluted to). This original concentration was determined after RNA extraction using the nanodrop. The elution volume of RNA was 50 µL, and the proportion of bee material used was ¼ of a bee per sample, thus we need to multiply by 4 to obtain the genome equivalents of 1 bee.

The maximum number of cycles for this assay was 35 cycles (equating to a critical threshold value = 30), above this non-specific and background cross-contamination could be detected additionally samples containing less than 100 copies of RNA were out of the range of quantification [28]. As a result, samples with PCR values less than 100 copies or with a critical threshold value of 30 or above were not included in further analysis. As the data did not follow a normal distribution, even after log^10^ transformation, the median and interquartile range of DWV-A and DWV-B genome equivalents was determined for each island. For the apiaries, the percentage of DWV-A *RdRp* and DWV-B *RdRp* was calculated using the genome equivalents. The percentages of colonies were then averaged to obtain the average for the apiary. The median and interquartile range of pupal samples were determined separately from the adult bees of Oahu island due to high variability. The medians were used to calculate the percentage of DWV-A and DWV-B on each island.

A Mann–Whitney U test was used to compare the viral loads (genome equivalents) on Oahu and Big Island. Kauai samples were excluded from this analysis as there were only 4 samples with quantifiable levels of DWV. Fisher’s exact probability tests were conducted to compare the prevalence of detectable and quantifiable amounts of DWV-A and DWV-B between the islands. The level of significance for all tests was *p* < 0.05.

### 2.6. Treated vs. Untreated Colonies

Out of the 41 colonies on Oahu, 15 were from managed apiaries that used *Varroa* treatment, and 15 were from managed apiaries that chose not to treat for *Varroa* mites. The remaining 11 colonies were feral colonies that did not receive treatment. The colonies were divided into the 3 groups to compare the differences in DWV-A and DWV-B load between them. The genome equivalents were log^10^ transformed and then tested for normality using the Ryan-joiner normality test and histogram plots. The data were normal, and thus student t-tests were used to look for significant differences in viral loads.

## 3. Results

### 3.1. Prevalence and Viral Titre

On the *Varroa*-free island of Kauai, DWV-A and -B were detected in 36% (8/22 colonies) and 59% (13/22 colonies) of colonies, respectively. However, the viral genome equivalents were only just quantifiable in four colonies, and these were low (10^5^ to 10^6^) (Table 1, Figure 1). In contrast, on the *Varroa*-infested islands of Oahu and Big Island, median DWV genome equivalents were several orders of magnitude greater ×10⁹). The levels of DWV-A on Oahu were not significantly different from the levels of DWV-A on Big Island (U = 809.5, *p* = 0.78), this was also the case for DWV-B (U = 692, *p* = 0.30).

Additionally, DWV-A and –B were detected in 100% of mite-infested colonies sampled on both islands (Oahu *n* = 41, Big Island *n* = 43) that was significantly greater than the number of colonies with detectable DWV-A (both *p* < 0.01) and DWV-B on Kauai (both *p* < 0.01). DWV-A and -B were also detected above the quantifiable threshold in 100% of colonies on Oahu and over 90% of colonies on Big Island (90.7% DWV-A 39/43 colonies, 95.3% DWV-B 41/43 colonies). The differences in the number of colonies with quantifiable DWV-A and DWV-B between Oahu and Big Island were not significant (DWV-A: *p* = 0.12 and DWV-B *p* = 0.49). However, both Oahu and Big Island had significantly more quantifiable cases of DWV-A (both *p* < 0.01) and DWV-B (both *p* < 0.01) than Kauai

The island genome equivalents of DWV-A vs. -B were not significantly different on Oahu (U = 793, *p* = 0.35) or Big Island (U = 713, *p* = 0.41), with DWV-A making up 46% and 59% of median genome equivalents on Oahu and Big Island, respectively. All of the Oahu pupal samples had quantifiable levels of DWV-B, but only 60% had quantifiable amounts of DWV-A, and 9 of the 10 samples were dominated by DWV-B (Table 1). Conversely, on Kauai, DWV-A and B co-infection were rarer, occurring in only 18% of colonies, and where coinfection occurred, only one variant was dominant whilst the other was below the quantifiable limit. For colony-level data, see Appendix A. All reported negative samples tested were negative of any DWV variant.

### 3.2. Treated vs. Untreated Colonies

All the colonies in each group, managed treated (*n* = 15), managed not-treated (*n* = 15), and feral (*n* = 11) had quantifiable amounts of DWV-A and DWV-B. For each group, the genome equivalents of DWV-B were not significantly different, i.e., feral vs. managed untreated (T = −0.43, *p* = 0.67), feral vs. managed treated (T = −1.59, *p* = 0.13) and managed untreated vs. managed treated (T = −1.13, *p* = 0.27). However, the DWV-A load was significantly lower in feral colonies than in managed, untreated colonies (T = −2.41, *p* = 0.027) or in feral than in managed, treated colonies (T = −2.16, *p* = 0.042). Managed treated and managed untreated colonies had similar levels of DWV-A (T = 0.04, *p* = 0.97) (Appendix A).

## 4. Discussions

In the original 2010 Hawaii study [9], the islands with *Varroa*, Oahu and Big Island, were entirely made up of the same DWV-A sequence. Our results indicate a large proportion of *RdRp* sequences now contain those that match the DWV-B variant. This suggests that the Hawaiian Islands of Oahu and Big Island are transitioning from DWV-A to DWV-B dominance, mirroring that observed in the UK, USA, Europe, South Africa (Figure 2) [16,20,28,29,30,31,32]. However, to confirm this would require future studies analyzing the full genome sequence of past and present samples from each island. Due to roughly a 100-fold increase in sensitivity of the PCR method [28], the viral genome equivalents in this study are not directly comparable to the original study. However, the relative ratios show that on Big Island and Oahu DWV-A is no longer solely dominant, and that DWV load on Kauai remains very low with a significantly lower prevalence of infected colonies compared to the two *Varroa* infested islands. In fact, on both Big Island and Oahu, the proportions of DWV-A and DWV-B are close to co-dominance, with DWV-A variants making up 59% and 46% of median genome equivalents on Big Island and Oahu, respectively. Additionally, at the colony level, 59% of colonies on Oahu are dominated by DWV-A and 56% on Big Island.

Intriguingly, the majority of change on Oahu appears to have occurred within the last three years, with samples from 2015 to 2016 consisting of mostly DWV-A (99% of reads) [26]. This is interesting because given the changing from DWV-A to DWV-B dominance overtime in other countries, one would expect the island which had hosted *Varroa* the longest, Oahu, to become dominated by DWV-B and to do so first. Whereas, it appears Big Island has become dominated more rapidly, with one study finding DWV-B domination in 2012 (96% of RNAseq reads) [26] and another in 2016 (>99% of RNAseq reads) [29] (Figure 3). However, whilst striking, these results should be interpreted with caution as coming from just 1 and 2 samples, respectively, they are not fully representative of the island at the time. In addition, it is fair to say that the change from DWV-A and DWV-B is not necessarily universal because, in South America, which was invaded by the mite some 50 years ago, DWV-A still prevails as the dominant variant (Figure 3) [37,38,45]. In fact, Ref. [37] only detected DWV-B within 3 of their 27 honey bee samples from Brazil. Whereas, in South Africa, DWV-B appeared to dominate from the mite’s introduction in 1997 or shortly afterward [30]. The median viral genome equivalent of DWV-A is similar on Oahu and Big Island, but the median viral genome equivalent of DWV-B on Oahu is half the value on Big Island (Table 1). A potential key difference between the colonies sampled was that the majority of the Big Island colonies were acaricide treated, whereas on Oahu, the colonies were a mix of treated, not treated, and feral (also not treated) colonies. Feral colonies that are able to resist *Varroa* without treatment had significantly lower DWV-A genome equivalents than treated colonies (Appendix A) but similar DWV-B genome equivalents. The managed untreated colonies had similar DWV-A levels to treated colonies, which were also significantly greater than levels in feral colonies. This is unexpected as other studies using the same methodology have found a reduced DWV burden in resistant, not treated, managed populations in South Africa and Brazil [30].

As expected, given the inefficiency of bee-to-bee routes of transmission [8], the number of DWV genome equivalents on the *Varroa* free island Kauai are still very low. Indeed, only four colonies had sufficient genome equivalents that were quantifiable. Additionally, in contrast to the original study, which detected DWV in 13% of colonies on *Varroa* free islands, we detected DWV in the majority of colonies on Kauai 77%. This result is attributed to the increased sensitivity of the methods used.

Recombinants been found to be prevalent within samples from Oahu and Big Island [26]. Considering the high incidence of co-infection, we found it is entirely possible that our samples from Big Island and Oahu could contain recombinants. However, as the RT-qPCR used in this study focused upon the *RdRp* region, we can only speculate on this possibility. Although the *RdRp* region is conserved and not known to be a common site for recombination relative to other regions of the genome [23,26,46].

Ultimately, this study has shown that since 2010 when DWV-B was not detected, the viral load and prevalence of DWV-B have increased to the point at which DWV-B now dominates colonies found on Big Island and co-dominates with DWV-A on Oahu. Thus far, this increase in DWV-B fits with what has been observed in numerous other regions (Figure 2) [16,20,30]. We know that DWV-B replicates to greater titers than DWV-A when injected into pupae [47,48] whilst being equally [48] or less virulent [49]. Furthermore, evidence suggests that DWV-B is able to replicate in *Varroa* mites, whereas DWV-A is not [18,50]. These findings help explain the field observations where DWV-B consistently occurs at higher titers than DWV-A [16]. The enhanced replication combined with a reduction in pupal virulence will give DWV-B the competitive edge during co-infection with DWV-A [50] since the 10–20% mortality of pupal infected with DWV-A prevents the vector (mites) from reproducing, hence breaking the transmission cycle. This may be negated by the fact that DWV-B is more virulent than DWV-A to caged adult bees [46], however, it seems unlikely as, especially in cases of high infestation, where irrespective of DWV variant colonies still collapse.

Additionally, it is curious, given the advantageous replicative abilities of DWV-B, why DWV-A initially gained dominance after *Varroa* spread to Oahu and Big Island. The reasons for this are at this point unclear, however, it has been shown that the rise of the near clonal master-variant (now called DWV-A) occurred within the pupae not the mite [14]. Once this occurred, either DWV-A was selected again in the pupae or more likely transmitted directly by *Varroa*. Perhaps the initial dominance is dependent on the variants present before *Varroa.* Between 1998 and 2009 484 mite and honey bee samples from 32 geographic regions testing positive for DWV, 83% were DWV-A, and the few DWV-B samples all originated from Europe [34]. Thus, perhaps DWV-B would have the chance to dominate if mites were to infest the island of Kauai.

Nonetheless, at this point, it is difficult to speculate at the future as there are still many gaps in our knowledge of the current prevalence of DWV-A and B worldwide that need to be filled (Figure 2). Indeed, it is not clear whether the two variants will continue to co-exist in Hawaii or whether DWV-B will eventually dominate Oahu and Big Island.

## Figures and Tables

**Figure 1 viruses-13-00969-f001:**
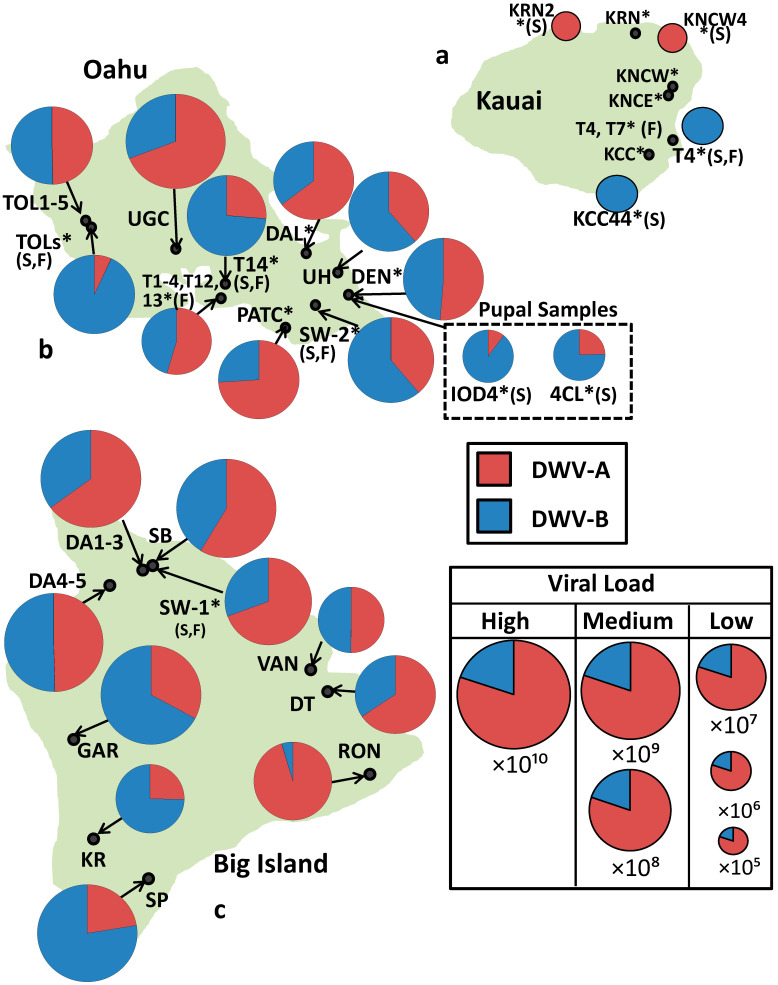
(**a**–**c**). Islands showing proportions of DWV-A *RdRp* (red) and DWV-B *RdRp* (blue) in each apiary (* = A colony that is not chemically treated for *Varroa*, S = Sample(s) came from a single colony, F = feral). The size of each pie chart is relative to the median total DWV genome equivalents per apiary.

**Figure 2 viruses-13-00969-f002:**
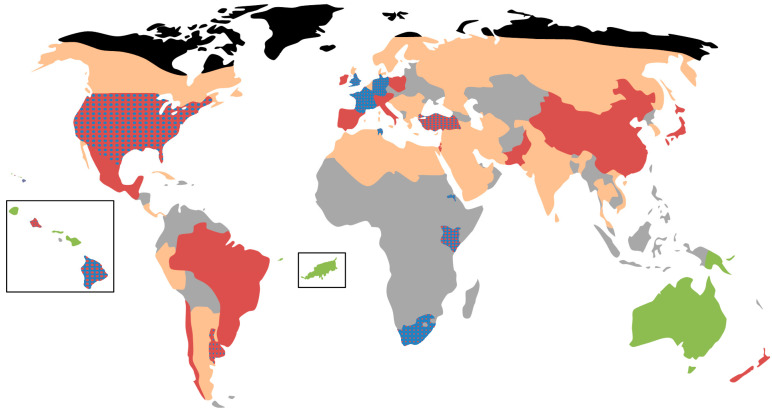
Global distribution of DWV in *Apis mellifera.* Red = DWV-A, blue = DWV-B, orange = DWV present but dominant strain unknown, grey = no data available, green = DWV absent or present at very low genome equivalents, Black = *Apis mellifera* absent. Blue dots on a red background indicate that DWV-A is dominant, but DWV-B is present conversely red dots on a blue background indicate that DWV-B is dominant, but DWV-A is present. The map was constructed by combining global level DWV data [33,34] with more detailed country level info as follows: Argentina (Buenos Aires and Sante Fe) [35], Australia [36], Brazil [37], Chile [38], China [39], Cuba [40], Ethiopia (Tigray) [41], Fernando de Noronha [25], France [20], Germany [32], Hawaii [This study, 26], Kenya [42], Papua new guinea [13], South Africa [30], Tunisia [43], Turkey [44], UK [16], Uruguay [45], USA [16]. The studies used to create this diagram were not required to have used the same primer set as our study.

**Figure 3 viruses-13-00969-f003:**
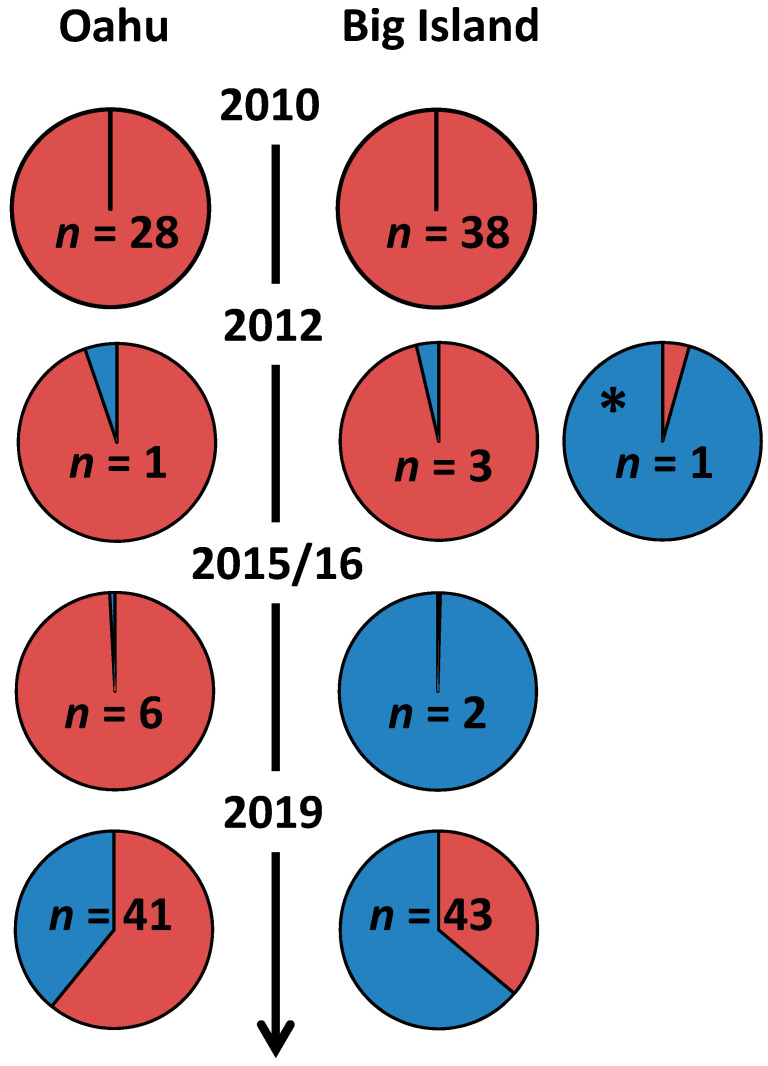
Changing proportions of DWV-A (red) and DWV-B (blue) on Big Island and Oahu over time. Sample sizes of the studies are given within the pie charts. Data for 2010 is from [9], 2012 [29], 2012 * [17], 2015/16 [26] and 2019 (this study). 2012 and 2012 * could not be combined due to the different methodologies used. N.B. Pie chart sizes do not convey DWV genome equivalents.

**Table 1 viruses-13-00969-t001:** Island median DWV genome equivalent and interquartile range (standard range for Kauai) and the year *Varroa* was first detected on each island.

Island	DWV-A	IQR	DWV-B	IQR
Kauai *Varroa*-free	7.53 × 10⁵ (*n* = 2)	2.07 × 10⁵	4.39 × 10⁶ (*n* = 2)	6.21 × 10⁶
Oahu Infested since 2007	1.03 × 10⁹ (*n* = 41)	1.69 × 10⁶	7.10 × 10⁸ (*n* = 41)	1.31 × 10⁶
Oahu—Pupae	1.44 × 10⁶ (*n* = 6)	5.13 × 10⁹	1.01 × 10⁷ (*n* = 10)	7.54 × 10⁶
Big island Infested since 2009	1.61 × 10⁹ (*n* = 41)	1.18 × 10^10^	1.42 × 10⁹ (*n* = 39)	2.32 × 10^10^

## Data Availability

The data presented in this study are available in the supplementary material Appendix A.

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
