# Peer review of "Ten Years of Deformed Wing Virus (DWV) in Hawaiian Honey Bees (Apis mellifera), the Dominant DWV-A Variant Is Potentially Being Replaced by Variants with a DWV-B Coding Sequence"

_viruses, 2021, doi:10.3390/v13060969_

Round 1

Reviewer 1 Report

Grindrod et al reported that DWV-B starts to dominate the honey bee colonies in Oahu and Big Island although DWV-A was once prevalent at a few years after the arrival of Varroa mites. The paper is quite descriptive but their observation would be valuable to understand the potential mechanism of shift in DWV variants while the mites remain infesting honey bee colonies. Several points the authors need to address or discuss are summarized below.

  1. Are there any data to show the infestation level of mites in the tested apiaries during 2010-2019?

  1. It would be very interesting to know if there has been the shift of DWV variant in the MITEs as well during 2010-2019.

  1. In ref. 9, it is not clear which variant was prevalent in Oahu and Big Island BEFORE the arrival of mites. Assuming it was similar to that of Kauai, why DWV-A became prevalent first AFTER the mite arrival? This should be at least discussed in the paper.

Author Response

  1. Are there any data to show the infestation level of mites in the tested apiaries during 2010-2019?

 Mite infestation data was not collected during 2010 nor 2019 studies, since either the majority colonies (expect the feral ones) were treated regularly so the mite infestation levels would only reflect the time since last treatment, or the mite free colonies on Kauai and Maui clearly had no mites.

2. It would be very interesting to know if there has been the shift of DWV variant in the MITEs as well during 2010-2019.

 This would be an interesting project, but most of the evidence suggests the main DWV variant in the bees is also in the mites (Wu, Dong, & Kadowaki, 2017), except during selection of a single master-variant which occurs in the pupae not the mite (Ryabov et al., 2014), hence we focused both studies on the honeybee not the mite. Furthermore, very recently it has been shown that DWV-A cannot replicate in Varroa but DWV-B can (Gisder & Genersch, 2021) – as now mentioned in introduction and at the end of the discussion

3. In ref. 9, it is not clear which variant was prevalent in Oahu and Big Island BEFORE the arrival of mites. Assuming it was similar to that of Kauai, why DWV-A became prevalent first AFTER the mite arrival? This should be at least discussed in the paper.

Ref 9 shows that on Big Island  when sampled in 2009 only the east side of the island had just become infected (see Fig. 1) we re-sampled 1 yr later in 2010 when the mite had spread throughout Big Island. In ref 9, supp data Fig. S2 shows in 2009 when most colonies where mite-free the strain diversity was large on Big Island as found on the Varroa-free islands of Maui and Kauai. In 2010 the variant diversity was greatly reduced Fig. S2. Thus, we can say with a high degree of certainty that pre-Varroa DWV strain diversity was large at least on Big Island. On Oahu, we have no pre-varroa samples so impossible to say if the same situation was found, but on two other islands where varroa in either absent or not a problem (Colonsay in Scotland and Fernando de Noronha in Brazil) both also have very low loads of highly diverse DWV variants, so we expect Oahu to been the same.     

We know that the rise of a near clonal master-variant (now called DWV-A) occurs in the pupae not the mite (Ryabov et al., 2014). Once this has occurred either DWV-A is selected again in the pupae or more likely transmitted directly by Varroa. This is evidenced by Wilfert et al., (2016) who analysed 389 honey bee and 95 mite samples from 32 geographic regions between 1998 and 2009 of those that tested positive for DWV, 83% were DWV-A and the few DWV-B samples all originated from Europe (see supp data Wilfert et al., 2016). We have therefore added a new section discussing this at the end of the discussion.

Reviewer 2 Report

Dear Authors,

Despite the interest in DWV evolution studies, your results obtained only by quantitative PCR, targeting the RdRp coding sequence of the DWV genome, provide only limited information on DWV-genotype changes in Hawaii Islands. In this manuscript, you can only conclude on the evolution of a small coding sequence, which is not representative of all the evolutions affecting the genome of DWV-recombinants. The sequencing of whole genome of the detected DWVs would better support your conclusions.

Analysis of genome equivalent loads is inappropriate because you calculated averages, standard errors, and you compared data that does not follow a normal distribution. I am assuming (because it is not clearly describe) for your quantitative PCRs you used a calibration curve with 10-fold dilutions, so you need to analyse and to compare the log10 values.

No statistical tests are described to support your results and conclusions. In addition, the conclusions are not supported by a rigorous presentation of the results.

Specific comments are reported on the PDF file.

Reviewer 3 Report

The manuscript entitled "Ten Years of Deformed Wing Virus (DWV) in Hawaiian Honey Bees (Apis mellifera), the Dominant DWV-A Variant Is Being Replaced by the DWV-B Variant", describes a recent virus screening of two variants of a critical honey bee pathogen, Deformed Wing Virus. Moreover, the authors use this new information about the DWV variants prevalence in Hawaii islands to identify an exciting shift from the variant DWV-A to the newly emerged and described more dangerous DWV-B from previous surveys from the past.

The authors performed an excellent review of the literature to support the conclusions and discuss the matter throughout the manuscript. The article is concise, well written, and a relevant topic in the field. It is the opinion of this reviewer that the manuscript is suitable for publication after addressing the following.

I would like to see in the manuscript a better explanation of why the authors eliminated honey bees with varroa mites for the survey. (Line 88)

Is this procedure used to eliminate the potential RNA extraction of Varroa mites tissues that could bias the study to the DWV-B variant, or were you looking specifically for bees that Varroa has not parasitized?

Author Response

I would like to see in the manuscript a better explanation of why the authors eliminated honey bees with varroa mites for the survey. (Line 88) 

Is this procedure used to eliminate the potential RNA extraction of Varroa mites tissues that could bias the study to the DWV-B variant, or were you looking specifically for bees that Varroa has not parasitized? –

Varroa mites were removed, if present, from honey bees to prevent any contamination of the samples with Varroa viral RNA, which could artificially increase the viral load. This has been clarified in the text. As previous mention in response to reviewer 1 the main DWV variant in the bees is also in the mites (eg Wu, Dong, & Kadowaki, 2017). Furthermore, if is not possible when sampling adults directly from the colony where, when and how they became infected with DWV, i.e. via mite feeding, or via natural transmission routes.

Round 2

Reviewer 1 Report

The authors did their best to address my comments on the paper in the revised version.

Author Response

Thanks, for the useful comments

Reviewer 2 Report

Thank you for your response and corrections. 

My last comments on the PDF file.

Author Response

L2: The title is confusing. You did not survey the evolution of the DWV for ten years. Only samples collected in november 2019 were analysed. Moreover, the shift from DWV-A to DWV-B was only accessed by RT-qPCR focusing the RdRp coding sequence. No partial or complete genome sequencing support your statment. The title should reflect the work realy performed. Please, can you modify the title?

Ten years of deformed wing virus refers to the fact that the first study on the DWV in Hawaiian honey bees (Martin et al., 2012) was conducted 10 years prior to the samples collected for this study.

However, we agree that the second part of the title needs correcting as we did not perform sequencing to confirm the presence of the variants. Therefore changed it to:

“Ten Years of Deformed Wing Virus (DWV) in Hawaiian Honey Bees (Apis mellifera), the Dominant DWV-A Variant Is Potentially Being Replaced by Variants with a DWV-B coding sequence”

L18: Please, can you modulate you conclusion taking into account that you performed limited experiments: RT-qPCRs amplifying the RdRp coding sequences for the typing of the three DWV master variants. No sequencing of the viral genomes was performed.

“we find that the DWV population has changed once again, with variants containing the RdRp coding sequence pertaining to the master variant B beginning to co-dominate alongside variants with the DWV-A RdRp sequence on the mite infested islands of Oahu and Big Island.”

L26: Two words for honey bees amended

L79: Please expand "RdRp". amended

L79: "coding sequences" instead of "genes" amended

L84: "coding sequence" amended

L146: A sapce beteween value and unit amended

L146: "RT-qPCR" instead of "RT q-PCR" amended

L147: I suggest the following clarification of the sentence: "RT-qPCR was performed on 50 ng of RNA sample (1 µl)..." amended

L149: The description of the RT-qPCR should be clarified. Could you briefly describe the method? A better description of the method has been added

“The 50 ng/µl samples were run singly alongside a 10-fold dilution series run in tripli-cate. The 10-fold dilution series was made using a standard specific to each DWV master variant, the concentration of which was determined using the Nanodrop 2000 (Thermo Fisher Scientific) before dilution. Reactions contained 1 µl of the 50 ng/µl RNA sample and 9 µl of master mix. The master mix was comprised of 0.08 µl reverse transcriptase, 1 µl DWV forward primer and 1 µl DWV reverse primer (Type A, B or C), 5 µl PCR mix and 1.92 µl Hâ‚‚0. A negative control consisting of 1 µl Hâ‚‚0 and 9 µl master mix was included on each PCR plate. An actin control was not deemed neces-sary as the samples had not undergone long term storage. The reactions were run on the quant studio 3, the reverse transcription stage occurred at 45 °C for 10 minutes and denaturation at 95 °C for 10 minutes, followed by 35 cycles of denaturation at 95 °C for 15 s, annealing at 58 °C (types A and B) or 61 °C (type C) for 15s and extension at 72 °C for 15 s. The final stage was a dissociation melt curve at 70 to 95 °C, this was to check for any contamination.” 

Did you use the actin control described by Kevill et al., 2017?

An actin control was not deemed necessary as the samples weren’t stored for a long time after extraction

How were produced the RNA?

RNA was extracted from the samples of crushed bees using the MagMAX mirVana total RNA isolation kit with the MagMAX express 96 as described in the RNA extraction and quantification section

What were they used for?

“To quantify the viral load of each DWV master variant, RT-qPCR was performed on the 116 samples using the ABC assay method [28]”.

How many replicates? Samples were run singly - “The 50 ng/µl samples were run singly”

The readers should understand what you performed without the help of the original description of the RT-qPCR.

L154: RdRp should be expanded in L79 amended

L161 : Should be described in RT-qPCR section. Amended

L168: Again, I do not understand this calculation. From which dilution is deduced the value of 50. Where do you discribe a 1/50 dilution of RNA sample? Is it the dilution you performed to adjust at 50 ng/µl? In such case, did not the dilutions depend of the samples (dilution factors reported in supplementary tables)? Can you clarify this dilution factor?

Sorry this was not clear, the dilution factor is the ratio between the initial concentration of the RNA which was determined using the nano drop and the concentration it was diluted to which was 50 µl/ng

The standardisation of the RNA was mentioned at the end of the section – RNA extraction and quantification

L182: "beyond" or "above"? amended

L186: 10 in Indice format amended

L195: What was the level of significance: p<0.05? Please indicate it. Amended

L220 and followings: I suggest you to express the p values with two significant decimals: p < 0.01 Amended

L293: I suggest you to link both sentances: "were similar: feral vs managed... Amended

L296: I suggest you to simpify: "or" instead of "and significantly lower in feral than in managed,". Amended

Supplementary figure 1 caption: Please can you describe the error bars. Amended

L347: The figure 2 is described after the figure 3 in the discussion section. Therefore, the figure n° should be 3 and not 2. Please, reorganize the figure numbers and dispositions. Agreed, the figures have been switched around